# Estimating and interpreting secondary attack risk: Binomial considered biased

Yushuf Sharker[1], Eben Kenah[2]*

**1** Division of Biometrics, Center for Drug Evaluation and Research, Food and Drug Administration, Silver Spring, Maryland, United States of America, **2** Biostatistics Division, College of Public Health, The Ohio State University, Columbus, Ohio, United States of America

* kenah.1@osu.edu

**Data Availability Statement:** All relevant data are within the manuscript and its Supporting information files.

**Funding:** EK and YS were supported by National Institute of Allergy and Infectious Diseases (NIAID)

## Abstract

The household secondary attack risk (SAR), often called the secondary attack rate or secondary infection risk, is the probability of infectious contact from an infectious household member $A$ to a given household member $B$, where we define infectious contact to be a contact sufficient to infect $B$ if he or she is susceptible. Estimation of the SAR is an important part of understanding and controlling the transmission of infectious diseases. In practice, it is most often estimated using binomial models such as logistic regression, which implicitly attribute all secondary infections in a household to the primary case. In the simplest case, the number of secondary infections in a household with $m$ susceptibles and a single primary case is modeled as a binomial($m, p$) random variable where $p$ is the SAR. Although it has long been understood that transmission within households is not binomial, it is thought that multiple generations of transmission can be neglected safely when $p$ is small. We use probability generating functions and simulations to show that this is a mistake. The proportion of susceptible household members infected can be substantially larger than the SAR even when $p$ is small. As a result, binomial estimates of the SAR are biased upward and their confidence intervals have poor coverage probabilities even if adjusted for clustering. Accurate point and interval estimates of the SAR can be obtained using longitudinal chain binomial models or pairwise survival analysis, which account for multiple generations of transmission within households, the ongoing risk of infection from outside the household, and incomplete follow-up. We illustrate the practical implications of these results in an analysis of household surveillance data collected by the Los Angeles County Department of Public Health during the 2009 influenza A (H1N1) pandemic.

## Author summary

The household secondary attack risk (SAR), often called the secondary attack rate or secondary infection risk, is the probability of infectious contact from an infectious household member $A$ to a given household member $B$, where we define infectious contact to be a contact sufficient to infect $B$ if he or she is susceptible. The most common statistical models used to estimate the SAR are binomial models such as logistic regression, which

grants R01 AI116770 and R03 AI124017. EK was also supported by National Institute of General Medical Sciences (NIGMS) grant U54 GM111274, and YS was also supported by National Institutes of Health (NIH) grant DP2 HD09179. NIAID: https://www.niaid.nih.gov/ NIGMS: https://www.nigms.nih.gov/ NIH: https://www.nih.gov/ The funders played no role in the study design, data collection and analysis, decision to publish, or the preparation of the manuscript.

**Competing interests:** The authors have declared that no competing interests exist.

implicitly assume that all secondary infections in a household are infected by the primary case. Here, we use analytical calculations and simulations to show that estimation of the SAR must account for multiple generations of transmission within households. As an example, we show that binomial models and statistical models that account for multiple generations of within-household transmission reach different conclusions about the household SAR for 2009 influenza A (H1N1) in Los Angeles County, with the latter models fitting the data better. In an epidemic, accurate estimation of the SAR allows rigorous evaluation of the effectiveness of public health interventions such as social distancing, prophylaxis or treatment, and vaccination.

## Introduction

In infectious disease epidemiology, the household secondary attack risk (SAR) is the probability of infectious contact from an infected household member $A$ to a susceptible household member $B$ during $A$'s infectious period, where we define infectious contact as a contact sufficient to infect $B$ if he or she is susceptible. It is often called the secondary attack *rate*, but we prefer to call it a *risk* because it is a probability [1]. SARs can also be defined in other groups of close contacts, such as schools or hospital wards [2].

The SAR is used to assess the transmissibility of disease and to evaluate control measures [3–7]. The idea was originally developed by Charles V. Chapin in 1903 to study the transmission of diphtheria and scarlet fever, and it was extended to influenza, tuberculosis, and other infectious diseases by Wade Hampton Frost [8–10]. Household surveillance data from emerging infections is often used to estimate the SAR, including 1957 and 1968 pandemic influenza [11–13], meningococcal disease [2], pertussis [6], SARS coronavirus [14], seasonal influenza [15–18], rotavirus [19], 2009 pandemic influenza A (H1N1) [20–26], MERS coronavirus [27, 28], Ebola virus disease [29–31], norovirus [32, 33], hand-foot-and-mouth disease [34], cryptosporidium [35], measles [36], and COVID-19 [37, 38].

It has been understood that within-household transmission is not binomial since the work of En'ko in 1899 [39], Reed and Frost in 1928 [40], and Greenwood in 1931 [41]. The process is binomial only if the primary case (the first infected household member [42]) is the only possible source of infection for susceptible household members throughout his or her infectious period. However, binomial models continue to be used for the estimation of the SAR because it is thought that multiple generations of transmission within households can be neglected safely when the SAR is small. In its simplest form, this assumes that the number of secondary infections in a household with $m$ susceptible individuals and a single primary case is a binomial$(m, p)$ random variable, where $p$ is the household SAR. A given transmission path of length $k$ from a primary case $A$ to a given susceptible $B$ has probability $p^k$, which decays exponentially as $k$ increases. Up to and including the COVID-19 pandemic, the vast majority of studies of household transmission use a binomial model (often a logistic regression model) to estimate the household SAR [2, 6, 7, 9, 11–13, 19, 22–28, 30–32, 35–38]. A smaller number of studies have used explicit statistical models of transmission [15–18, 20, 21, 29, 33, 43]. Here, we hope to establish that the latter approach should become universal.

Although the probability of each given transmission path of length $k$ from $A$ to $B$ decays as $p^k$, the risk of infection through $k$ generations of transmission also depends on the number of possible paths of length $k$. A path of length $k \geq 1$ from $A$ to $B$ can be specified by choosing $k − 1$ individuals from the $m − 1$ susceptible household members other than $B$. Each ordering of these $k − 1$ individuals produces a unique transmission path. For $1 \leq k \leq m$, the total

number of paths from $A$ to $B$ of length $k$ equals the number of permutations of $k - 1$ objects chosen from $m - 1$ objects:

$$P(m - 1, k - 1) = \frac{(m - 1)!}{(m - k)!}.\tag{1}$$

Table 1 shows that the number of paths of length $k$ can grow quickly with household size. Each path can carry infection from $A$ to $B$, so the total risk of transmission from $A$ to $B$ along any path of length $k$ can be much greater than $p^k$. A binomial model attributes this additional risk of infection to direct transmission from the primary case, so the estimated SAR is too high.

The binomial variance assumes that infections in different household members are independent. Because each new infection in a household increases the risk of infection in the remaining susceptibles, infections within a household are positively correlated. This correlation makes the true variance in the number of infections larger than the binomial variance. To address this issue, cluster-adjusted variances [6, 21, 23, 24, 31] and random effects [32] have been used to account for correlation among household members. Because of the bias in the point estimate of the SAR, this adjustment for clustering does not produce confidence intervals that have the expected coverage probabilities.

When the latent period (between infection and the onset of infectiousness) and incubation period (between infection and the onset of symptoms) are longer than the infectious period, generations of infection can be separated in time. This was seen seen most famously by Peter Panum in a measles epidemic on the Faroe Islands in 1846 [44]. With such separation, a binomial model could be used to estimate the risk of infection within a follow-up interval designed to capture only the first generation of transmission. However, such separation of generations is unusual. For example, the incubation period of influenza is roughly 1–2 days and the duration of viral shedding is 4–6 days [45, 46]. For influenza and most other infectious diseases, the follow-up times of households cannot be adjusted to capture exactly one generation of transmission.

In its original usage, the SAR was defined as the probability that a susceptible in a household with a primary case is infected by within-household transmission, whether or not there were multiple generations of transmission within the household [8, 9]. Here, we will call this the household final attack risk (FAR). With complete follow-up of all households, a cluster-adjusted binomial model could produce an unbiased estimate of the FAR. However, the estimated FAR will be biased upward if there are co-primary cases or if household members are at risk of infection from outside the household during the follow-up period [3, 9, 47]. Such conditions are common in practice.

In its modern interpretation, the household SAR is an extremely useful measure of the transmissibility of infection. However, this interpretation requires us to abandon the use of binomial models for estimation. Here, we use probability generating functions and simulations to show that (1) a binomial model produces biased estimates of the household SAR even when

**Table 1. Number of paths from the primary case to a given susceptible.**

| Susceptibles | Path length ($k$) | | | |
|---|---|---|---|---|
| ($m$) | 1 | 2 | 3 | 4 |
| 2 | 1 | 1 | 0 | 0 |
| 4 | 1 | 3 | 6 | 6 |
| 9 | 1 | 8 | 56 | 336 |

the probability of transmission is small and (2) cluster adjustment of the variances does not produce interval estimates with the expected coverage probabilities. To estimate the household SAR, explicit statistical models of disease transmission such as longitudinal chain binomial models [40, 48] or pairwise survival analysis [49–52] should always be used. We illustrate the practical implications of these results using household surveillance data collected by the Los Angeles County Department of Public Health during the 2009 influenza A (H1N1) pandemic. In these data, binomial models produce SAR estimates that are too high to be interpreted as probabilities of transmission.

## Methods

For simplicity, our analytical calculations and simulations assume a uniform SAR within households (i.e., no variation in infectiousness or susceptibility) and no risk of infection from outside the household except for the primary case. These assumptions are not realistic, but binomial models break down even under these ideal conditions. We use probability generating functions (PGFs) to calculate the true outbreak size distributions at different combinations of the number of susceptibles ($m$) and the SAR ($p$), and we verify these calculations in simulations of household outbreaks.

### Household outbreak size distributions

Assume that each infectious member of a household makes infectious contact with each other member of the household with probability $p$ during his or her infectious period. Let $p_{mi}$ be the probability that $i$ out of $m$ susceptibles are infected by within-household transmission in a household with a single primary case. Then

$$g_m(x) = \sum_{i=0}^{m} p_{mi} x^i \qquad (2)$$

is the probability generating function (PGF) for the outbreak size distribution in a household with $m$ susceptibles and one primary case. Because a household with zero susceptibles has zero secondary infections with probability one, $g_0(x) = 1$.

The PGF for the outbreak size distribution in a household with $m + 1$ susceptibles can be derived from the PGFs for smaller households. Imagine a household with $m$ susceptibles of whom $i$ were infected. Now imagine that the household had one more susceptible. There are two possible outcomes:

1. With probability $(1 - p)^{i+1}$, the additional susceptible escapes infection from all $i + 1$ infected household members. The total number of infections in the household is $i$.

2. With probability $1 - (1 - p)^{i+1}$, the additional susceptible gets infected. He or she acts like a primary case in a household containing the $m - i$ susceptibles who escaped infection. There are $i + 1$ infections, and the number of infections among the remaining susceptibles has the PGF $g_{m-i}(x)$.

Combining these results, we conclude that

$$g_{m+1}(x) = \sum_{i=0}^{m} p_{mi}[(1 - p)^{i+1} x^i + (1 - (1 - p)^{i+1}) x^{i+1} g_{m-i}(x)] \qquad (3)$$

The first few iterations yield

$$g_0(x) \quad = 1, \qquad (4)$$

$$g_1(x) \quad = (1-p) + px, \tag{5}$$

$$g_2(x) \quad = (1-p)^2 + 2p(1-p)^2 x + (3p^2 - 2p^3)x^2 \tag{6}$$

which can be checked by hand. We calculated these polynomials using Python code in S2 File. As shown in Eq (2), the coefficient on $x^i$ in the PGF $g_m(x)$ is the probability that $i$ of $m$ susceptibles are infected in a household outbreak started by a single primary case. Using these probabilities, we can calculate the mean and variance of the number of infections among the $m$ susceptibles.

## Household outbreak simulations

We simulated household outbreaks using Erdős-Rényi random graphs [53, 54], where each pair of nodes is connected independently with probability $p$. In our graphs, each node represents a household member and $p$ is the SAR. One node is fixed as the primary case, and all household members connected to the primary case by a series of edges are infected. This captures the final outcome of one realization of a within-household epidemic with SAR $p$. This simple model allows but does not require multiple generations of infection within households, allowing us to evaluate the performance of binomial estimates based on the assumption of a single generation of transmission.

We performed 40,000 simulations for each combination of household size and SAR. In each simulation, there were 200 independent households of the same size. We used logistic regression to calculate the proportion of susceptible household members who were infected with a naive 95% confidence interval. We then calculated a cluster-adjusted confidence interval using generalized estimating equations (GEE) with a robust variance estimate. The variance inflation factor (VIF) was calculated as the ratio of the robust variance to the naive variance. All confidence intervals were calculated on the logit scale as $\hat{\beta} \pm 1.96\,\hat{\sigma}$ where $\hat{\beta} = \text{logit}\,(\hat{p})$ is the estimated log odds of infection and $\hat{\sigma}$ is the naive or robust standard error estimate. Finally, we transformed the confidence intervals to the probability scale and estimated the coverage probabilities for the true household SAR and the true household FAR.

**Source code.** Simulations were implemented in Python 3 [55], and statistical analysis was performed in R [56]. The R code is available in S1 File, and the Python code is available in S2 File. All software used is free and open-source, and further details are given in the Supporting Information.

## Household data analysis

To give a practical example of the consequences of using a binomial model to estimate the household SAR, we use influenza A (H1N1) household surveillance data collected by the Los Angeles County Department of Public Health (LACDPH) between April 22 and May 19, 2009. The data was collected using the following protocol [49]:

1. Nasopharyngeal swabs and aspirates were taken from individuals who reported to the LACDPH or other health care providers with acute febrile respiratory illness (AFRI), defined as a fever $\geq 100°$F plus cough, core throat, or runny nose. These specimens were tested for influenza, and the age, gender, and symptom onset date of the AFRI patient were recorded.

2. Patients whose specimens tested positive for pandemic influenza A (H1N1) or for influenza A of undetermined subtype were enrolled as primary cases. Each of them was given a

structured phone interview to collect information about his or her household contacts. They were asked to report the symptom onset date of any AFRI episodes among their household contacts.

3. When necessary, a follow-up interview was given 14 days after the symptom onset date of the primary case to assess whether any additional AFRI episodes had occurred in the household, including their illness onset date.

For simplicity, we assume all AFRI episodes among household members were caused by influenza A (H1N1) and that all household members except the primary case were susceptible to infection. All analyses use natural history assumptions adapted from Ref [20] and consistent with Ref [46]. Identical assumptions were used in Refs [50, 51]. In the primary analysis, we assumed an incubation period of 2 days, a latent period of zero days, and an infectious period of 6 days. We also consider 4-day and 8-day infectious periods.

We estimated the household SAR for 2009 pandemic influenza A (H1N1) using binomial models, a longitudinal chain binomial model [40, 48], and parametric pairwise regression models [52, 57]. In each household, we censored observations at the end of the infectious period of the primary case. Thus, the models are fit only to infections that could have been caused by primary cases, giving the binomial models the best possible chance of accurately estimating the household SAR. For each assumed infectious period, all statistical models were fit to exactly the same data. For simplicity, we did not include any covariates in these analyses. Final size chain binomial models were not used because they require complete observation of each within-household epidemic, so they cannot be fit to data censored at the end of the infectious period of the primary case in each household.

**Binomial models.** Two binomial models were fit to the LACDPH households data. The first model was an intercept-only logistic regression model—a binomial generalized linear model (GLM) with logit link. The second model was an intercept-only binomial GEE model [58]. For both models, we calculated Wald confidence intervals using naive and cluster-adjusted variances [59].

**Longitudinal chain binomial model.** The chain binomial model assumes that a given infectious person $A$ makes infectious contact with a given susceptible household member $B$ with an unknown probability $p$ on each day that $A$ is infectious. On day $t$, an individual $B$ who is exposed to $k$ infectious household members will escape infection with probability $q^k$ and be infected with probability $1 - q^k$, where $q = 1 - p$. The likelihood contribution from observation of individual $B$ is the product of these likelihood contributions over all days where $B$ was at risk of infection. The overall likelihood is the product of the likelihood contributions of all susceptibles who were at risk of infection for at least one day.

The household SAR is $1 - q^\iota$ where $\iota$ is the infectious period. Because $p \in (0, 1)$, our likelihood was defined in terms of $\text{logit}(p) = \ln(p/q)$. To get a point estimate of the SAR, the unknown true $q$ is replaced by a point estimate $\hat{q} = 1 - \hat{p}$. Standard maximum likelihood estimation was used to get point and interval estimates on the logit scale, which were transformed back to the probability scale. For simplicity, we have assumed that the probability of escaping infection from an infectious household member does not depend on how long he or she has been infectious or on any covariates. More sophisticated longitudinal chain binomial models can allow the escape probability to vary with the time since infection or with covariates [40, 48].

**Pairwise survival analysis.** Pairwise survival analysis estimates failure times in ordered pairs consisting of an infectious individual and a susceptible household member [57]. The pair AB is at risk of transmission starting with the onset of infectiousness in A, and failure occurs if

A infects B. This failure time, called a *contact interval* is right-censored if B is infected by someone other than A or if observation of the pair stops. To account for uncertainty about who-infected-whom, the overall likelihood is the sum of the likelihoods for all possible combinations of who-infected-whom consistent with the data [49]. The survival function $S(\tau, \theta)$, where $\theta$ is a parameter vector, is the probability that the contact interval is greater than $\tau$. If $\theta_0$ is the true value of $\theta$ and the infectious period is $\iota$, then the household SAR is $1 - S(\iota, \theta_0)$. To get a point estimate of the SAR, the unknown true parameter $\theta_0$ is replaced by the maximum likelihood estimate $\hat{\theta}$.

We used intercept-only exponential, Weibull, and log-logistic regression models [52]. For the exponential distribution, $S(\tau, \lambda) = \exp(-\lambda\tau)$ where $\lambda$ is the rate parameter. For the Weibull distribution, $S(\tau, \lambda, \gamma) = \exp[-(\lambda\tau)^\gamma]$ where $\lambda$ is the rate and $\gamma$ is the shape parameter. For the log-logistic distribution, $S(\tau, \lambda, \gamma) = [1 + (\lambda\tau)^\gamma]^{-1}$ for rate $\lambda$ and shape $\gamma$. For all three distributions, $\lambda > 0$ and $\gamma > 0$ so we defined our likelihoods in terms of their natural logarithms $\ln\lambda$ and $\ln\gamma$. Standard maximum likelihood estimation was used to get point estimates and a covariance matrix for the rate and shape parameters. To get a 95% confidence interval for the SAR, we sampled $\ln\lambda$ and $\ln\gamma$ from their approximate multivariate normal distribution, calculated the household SAR for each sample, and took the 2.5% and 97.5% quantiles of the calculated SARs as confidence limits.

**Goodness of fit.**   To see how well the SAR estimates fit the data, we simulated outbreaks in the Los Angeles households using SAR point estimates from the binomial model, the chain binomial model, and pairwise survival models. In each simulation, we calculated the total number of infections among susceptible household members. For each SAR estimate, we performed 4,000 simulations. We then compared the simulated household epidemics to the observed final size of the outbreak started by the primary cases (i.e., the total number of cases who can be linked to a primary case through one or more generations of transmission). For all infectious periods shorter than 12 days, there are a few observed cases that occur after the end of the initial within-household outbreak. Given each assumed infectious period, these late cases are excluded because they can only be explained by later introductions of infection into the household or by transmission paths that include undetected cases.

**Source code.**   Statistical analyses were done with R [56], and the simulations were implemented in Python 3 [55]. The R code is available in S3 File, the Python code is available in S4 File, and the household data are available in S5 File. All software used is free and open-source, and further details are given in the Supporting Information.

## Results

### Household outbreak simulations

Fig 1 shows the household FAR calculated using PGFs (lines) and from simulations (symbols) as a function of the true SAR and the number of susceptibles. There is excellent agreement between the analytical calculations and the simulations. Both show that the household FAR is larger than the household SAR when there is more than one susceptible. At a fixed SAR, the difference between the SAR and the FAR increases with household size. Thus, a binomial model will produce a point estimate of the SAR that is biased upward whenever there is more than one susceptible household member.

Fig 2 shows the VIF calculated using PGFs (lines) and from simulations (symbols) as a function of the true SAR and the number of susceptibles. Again, there is excellent agreement between the analytical calculations and the simulations. The variance of the number of infections within households is substantially larger than the binomial variance, and this difference

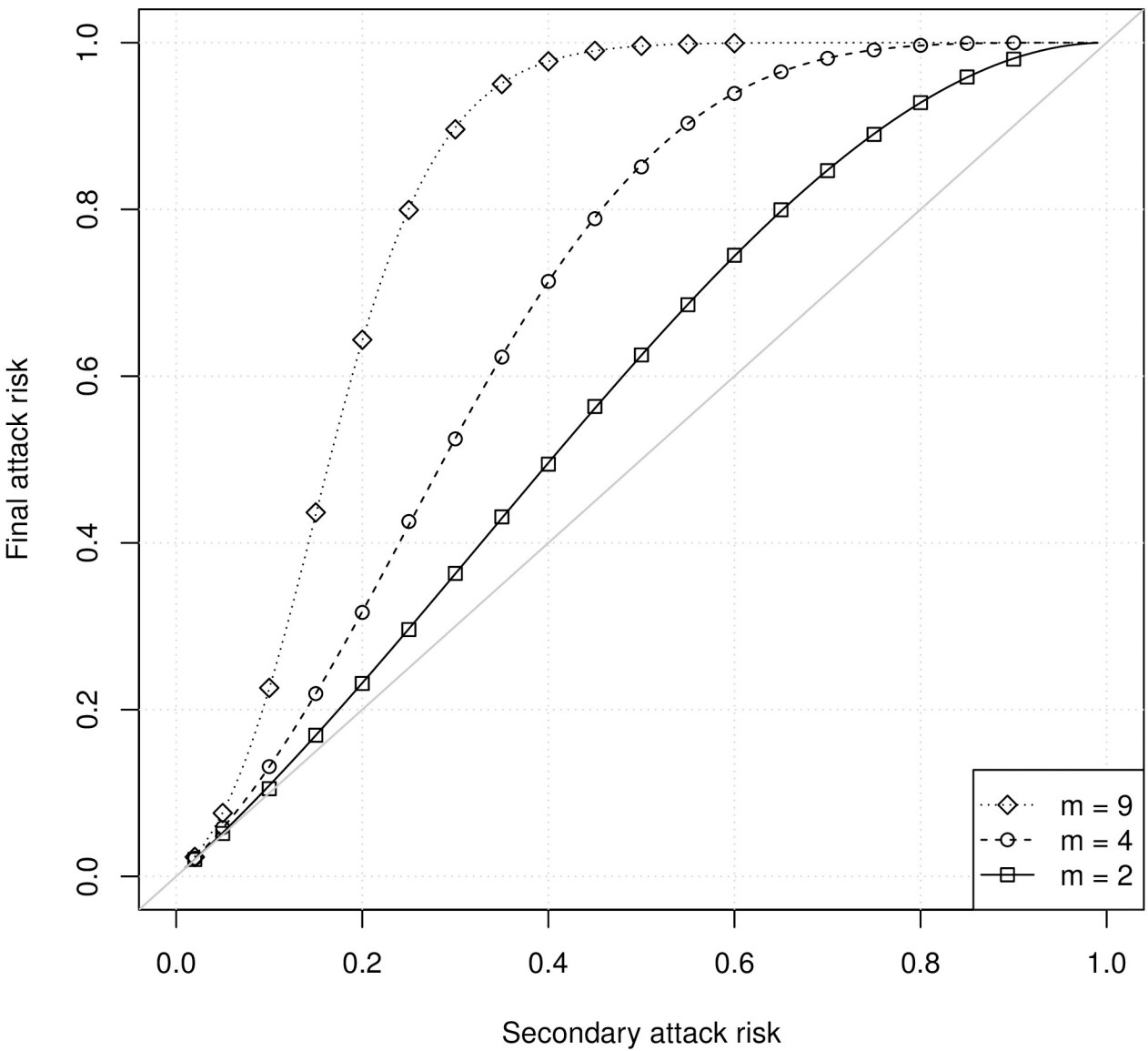

**Fig 1. The household FAR as a function of the SAR for households with different numbers of susceptibles *m*.** Lines show analytical calculations using probability generating functions, and simulations show estimates from 40,000 simulated household outbreaks. Each simulated household outbreak had a single primary case, so the total household size was *m* + 1.

increases with increasing household size. Thus, confidence intervals based on a binomial estimate will have coverage probabilities that are too low even if the estimated SAR is correct.

Fig 3 shows the household SAR coverage probabilities for unadjusted and cluster-adjusted binomial 95% confidence intervals. Even for small households, the coverage probabilities are below 95% and decrease rapidly as the true SAR increases. Cluster adjustment increases the coverage probabilities only slightly. With or without adjustment for clustering by household, a binomial model does not produce reliable point or interval estimates of the household SAR.

Fig 4 shows coverage probabilities of unadjusted and cluster-adjusted 95% confidence intervals for the household FAR. Coverage of the FAR is much higher than coverage of the SAR. However, the coverage probabilities for unadjusted confidence intervals are always below 95%, and they decrease with increasing household size or increasing SAR. Adjustment

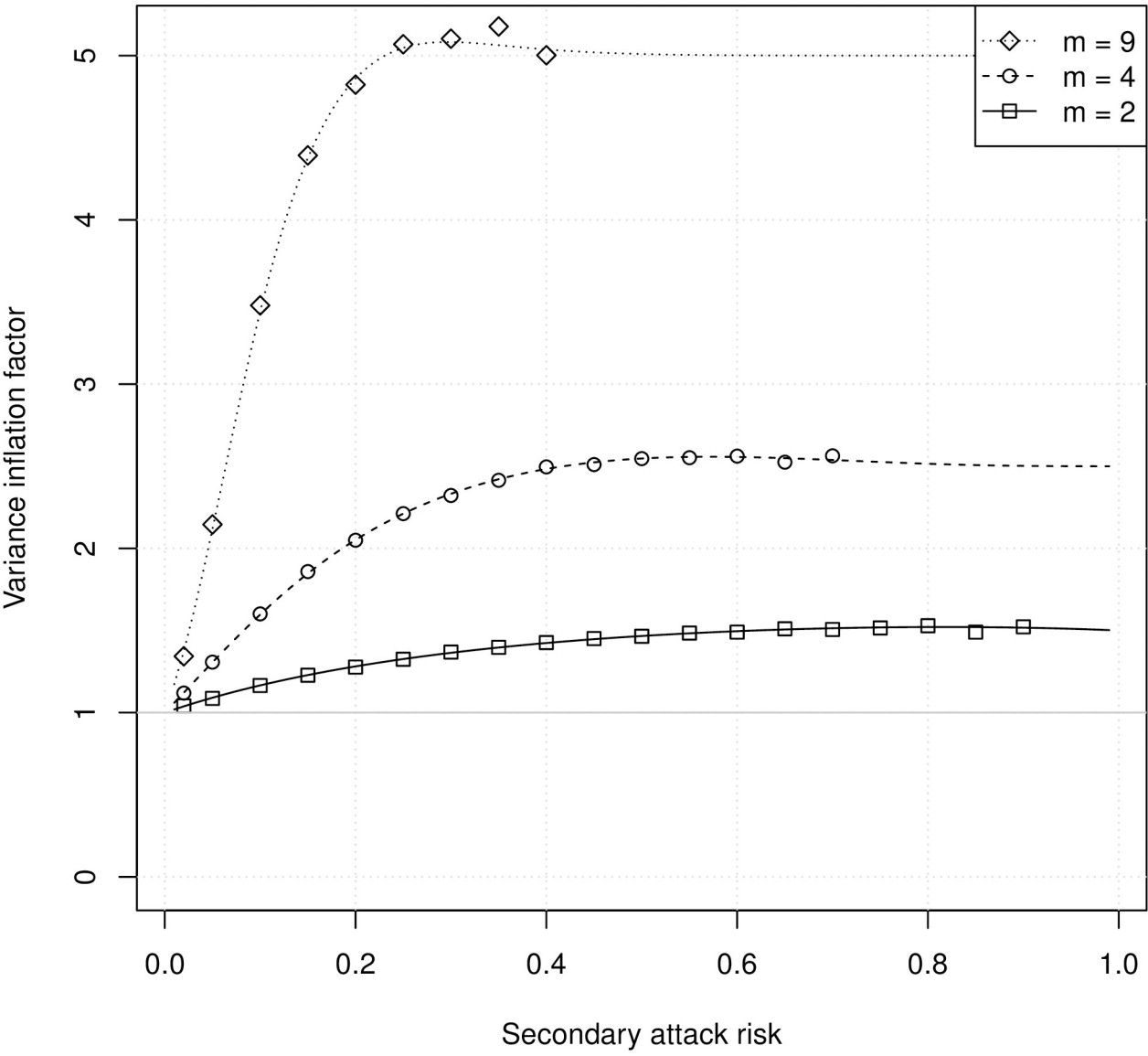

**Fig 2. The VIF as a function of the SAR for households with *m* susceptibles.** Lines show analytical calculations, and symbols show estimates from 40,000 simulated household outbreaks. Each simulated household outbreak started with a single primary case, so the total household size was *m* + 1. For numerical stability, symbols are shown only for simulations with an observed FAR <0.99.

for clustering by household corrects this problem, producing coverage probabilities close to 95% for all household sizes. Under these ideal conditions, a binomial model can produce reliable point and interval estimates of the household FAR when the variance is adjusted for clustering within households. This does not imply that FAR can be defined clearly or estimated accurately under more realistic conditions, and it does not imply that the FAR is an acceptable substitute for the SAR in practice.

## Household data analysis

In the LACDPH pandemic influenza A (H1N1) data, there were 58 households with a total of 299 members. There were 99 infections, of which 62 were classified as primary cases because 4

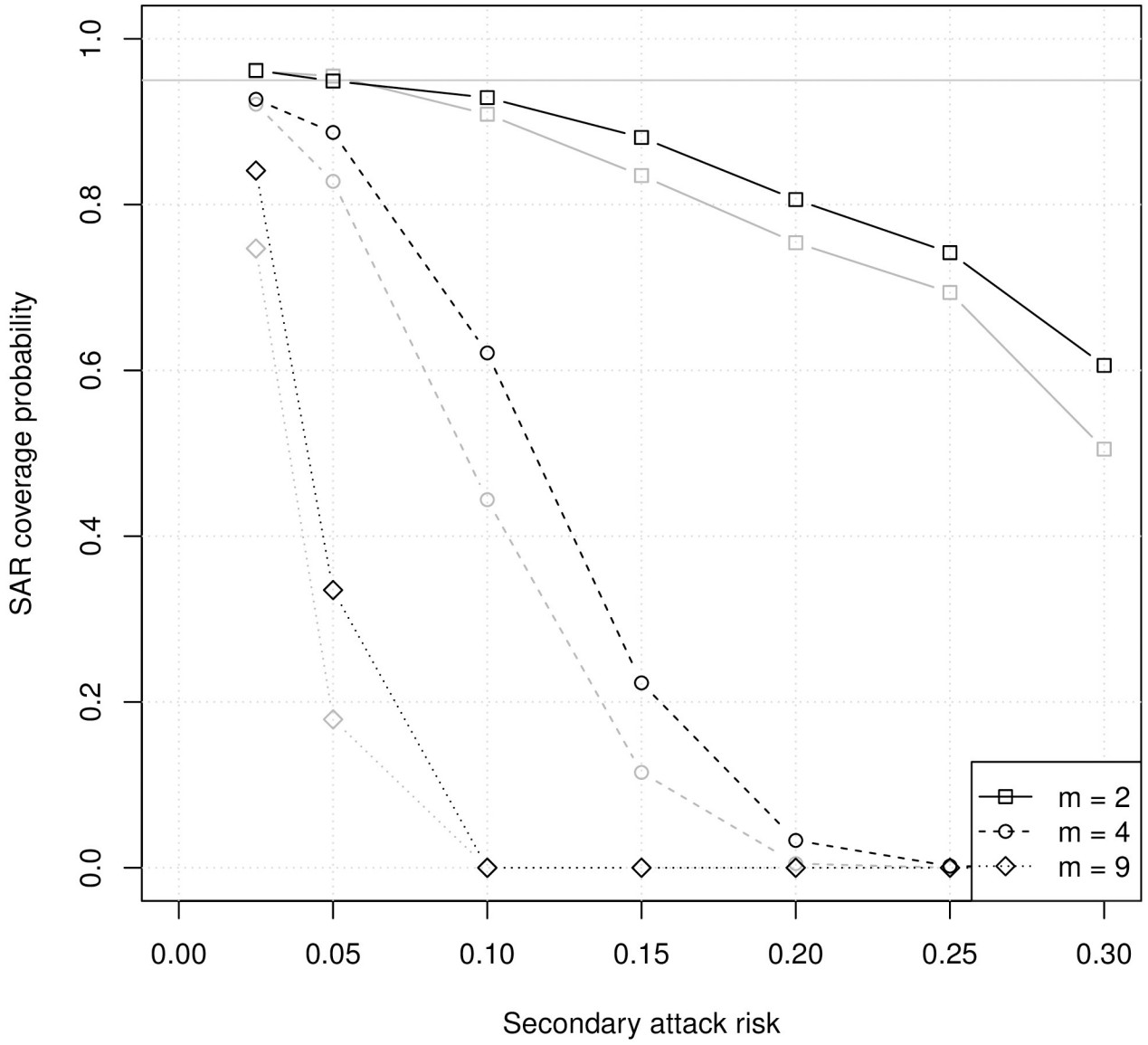

**Fig 3. Coverage probabilities of binomial 95% confidence intervals for the household SAR with different numbers of susceptibles (*m*).** Gray lines are coverage probabilities for unadjusted confidence intervals, and black lines are coverage probabilities for cluster-adjusted confidence intervals. Each symbol represents 1,000 simulations with 100 households each.

of 58 households had two co-primary cases. There were 37 household contacts who were infected while under observation. The median household size was 5 with a range from 2 to 20. Both in this example and more generally, co-primary cases and varying household sizes are practical problems for estimation of the household SAR.

There are three types of cases relevant to our analyses: *Possible second generation cases* are susceptible household members who are infected during the infectious period of the primary case, so it is possible that they were infected by the primary case. *Final size cases* are susceptible household members who could have been infected through a transmission path starting from a primary case. *Late cases* are susceptible household members who were infected after the end of the infectious period of the last final size case in the household. Given the assumed infectious period, these cases can only be explained by a new introduction of infection to the

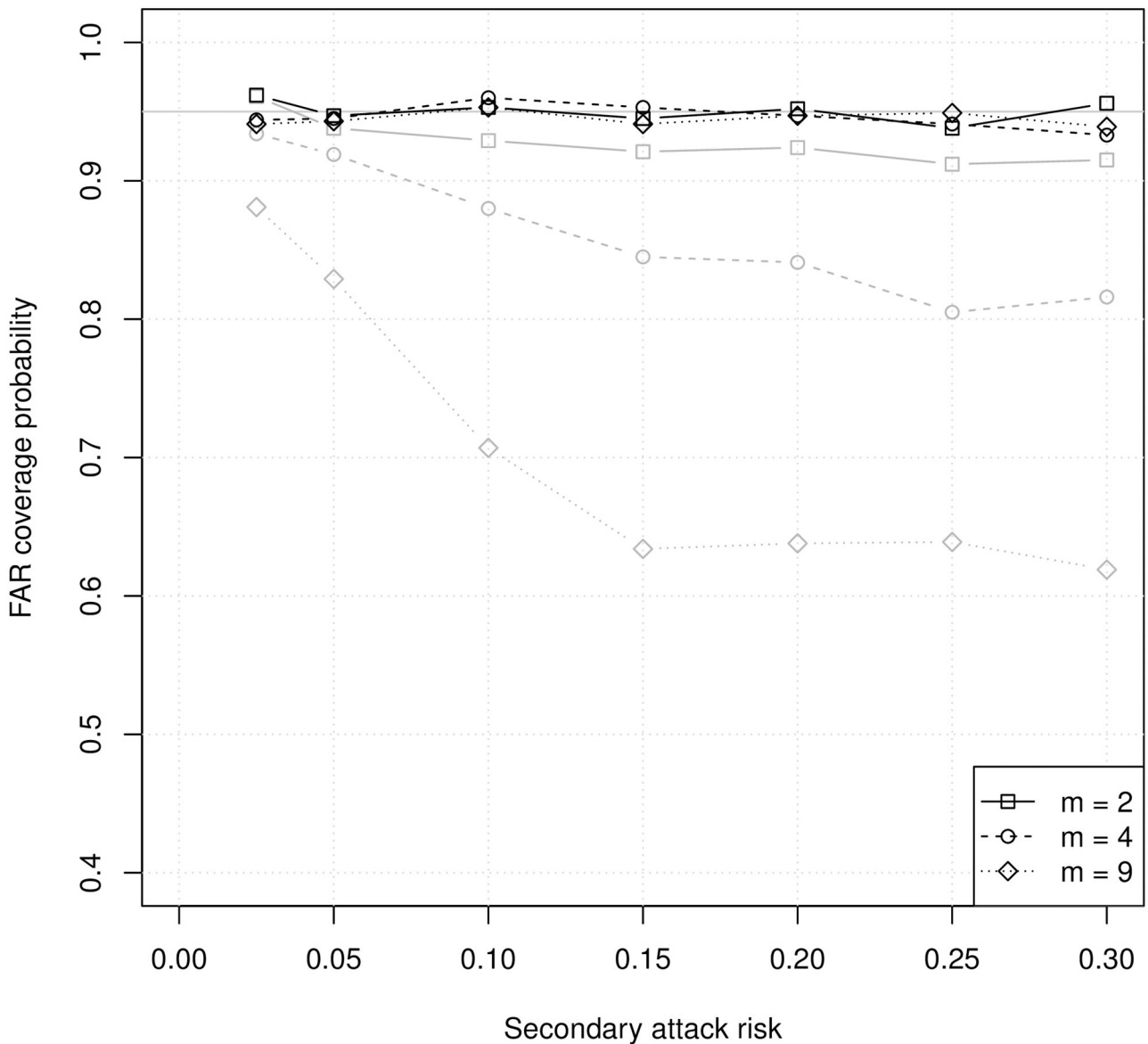

**Fig 4. Coverage probabilities of binomial 95% confidence intervals for the household FAR with different numbers of susceptibles ($m$).** Gray lines are coverage probabilities for unadjusted confidence intervals, and black lines are coverage probabilities for cluster-adjusted confidence intervals. Each symbol represents 1,000 simulations with 100 households each.

household or by transmission paths that include undetected cases. In volunteer challenge studies, approximately 71% of influenza A (H1N1) infections result in symptoms and 37% result in fever $\geq 100°$F [46]. In the analyses comparing binomial models to chain binomial and pairwise survival models, we use only the possible second-generation cases to give the binomial models the best possible chance of producing a good estimate of the household SAR.

Table 2 shows the numbers of possible second generation cases, final size cases, and late cases for each assumed infectious period from 3 days (probably too short) to 12 days (almost surely too long). Assuming an infectious period of 6 days as in our primary analysis, there are 24 possible second generation cases, 26 final size cases, and 11 late cases. We also show analyses with 4-day and 8-day infectious periods.

**Table 2. The number of possible second generation cases, final size cases, and late cases for each assumed infectious period.** There are always 37 total final size and late cases.

| Infectious period (days) | Possible second generation cases | Final size cases | Late cases |
|---|---|---|---|
| 3 | 13 | 16 | 21 |
| 4 | 17 | 22 | 15 |
| 5 | 20 | 25 | 12 |
| 6 | 24 | 26 | 11 |
| 7 | 28 | 32 | 5 |
| 8 | 28 | 32 | 5 |
| 9 | 28 | 32 | 5 |
| 10 | 32 | 36 | 1 |
| 11 | 33 | 36 | 1 |
| 12 | 34 | 37 | 0 |

Table 3 shows point estimates and 95% confidence intervals for the household SAR. The point estimates for all binomial models are identical. As expected, binomial models produce much higher estimates than the chain binomial or pairwise regression models. Adjustment for clustering produced wider confidence intervals, with very similar results in the binomial GLM

**Table 3. Estimates of the household SAR with 95% confidence limits and Akaike information criterion (AIC) for pairwise regression models.**

| Model | | Estimated SAR | | AIC |
|---|---|---|---|---|
| **6-day infectious period** | | | | |
| Binomial: | GLM (naive) | 0.101 | (0.067, 0.144) | |
| | GLM (adjusted) | 0.101 | (0.052, 0.189) | |
| | GEE (naive) | 0.101 | (0.069, 0.147) | |
| | GEE (robust) | 0.101 | (0.052, 0.188) | |
| Longitudinal chain binomial | | 0.076 | (0.051, 0.109) | |
| Pairwise regression: | exponential | 0.075 | (0.051, 0.110) | 235.96 |
| | Weibull | 0.079 | (0.056, 0.138) | 236.38 |
| | log-logistic | 0.079 | (0.055, 0.133) | 236.26 |
| **4-day infectious period** | | | | |
| Binomial: | GLM (naive) | 0.072 | (0.043, 0.109) | |
| | GLM (adjusted) | 0.072 | (0.033, 0.150) | |
| | GEE (naive) | 0.072 | (0.045, 0.112) | |
| | GEE (robust) | 0.072 | (0.033, 0.149) | |
| Longitudinal chain binomial | | 0.059 | (0.035, 0.090) | |
| Pairwise regression: | exponential | 0.058 | (0.036, 0.092) | 166.54 |
| | Weibull | 0.063 | (0.041, 0.135) | 162.67 |
| | log-logistic | 0.063 | (0.042, 0.133) | 162.65 |
| **8-day infectious period** | | | | |
| Binomial: | GLM (naive) | 0.118 | (0.081, 0.163) | |
| | GLM (adjusted) | 0.118 | (0.063, 0.211) | |
| | GEE (naive) | 0.118 | (0.083, 0.166) | |
| | GEE (robust) | 0.118 | (0.063, 0.210) | |
| Longitudinal chain binomial | | 0.085 | (0.058, 0.118) | |
| Pairwise regression: | exponential | 0.084 | (0.059, 0.120) | 281.88 |
| | Weibull | 0.085 | (0.062, 0.145) | 283.77 |
| | log-logistic | 0.085 | (0.062, 0.139) | 283.49 |

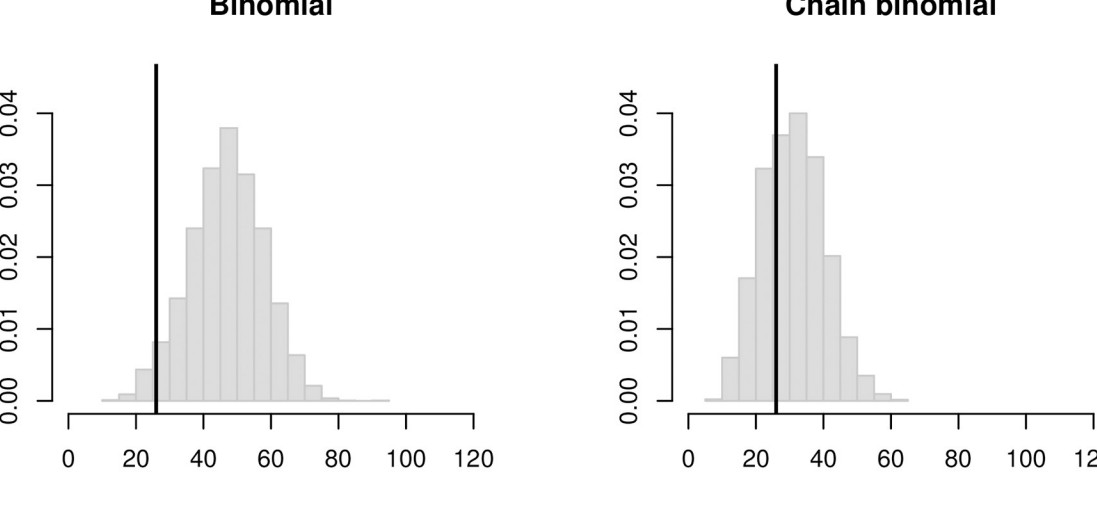

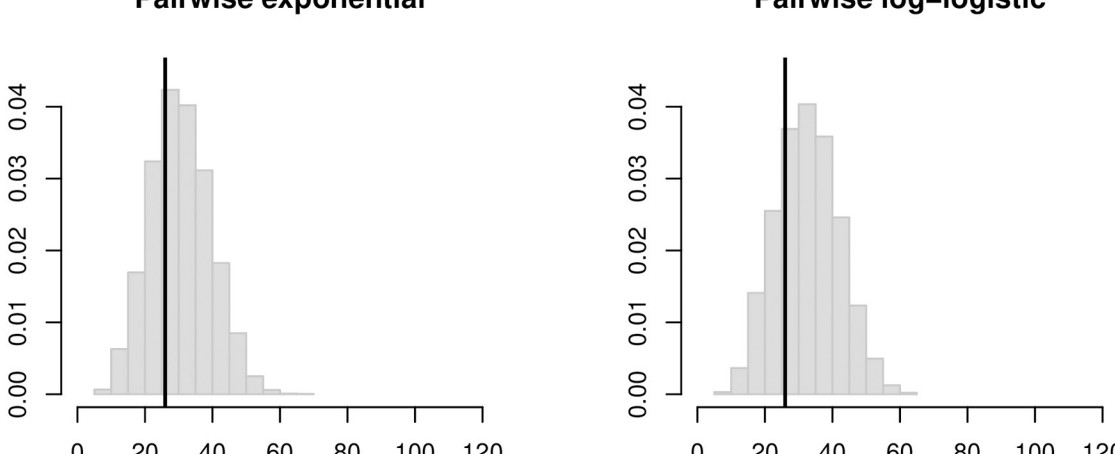

**Fig 5. Histograms of simulated final outbreak sizes in the LA households based on household SAR estimates assuming a 6-day infectious period.** Vertical black lines indicate the observed final size of 26 cases.

and GEE models. The chain binomial and exponential pairwise regression models produced nearly identical point and interval estimates of the household SAR. For each assumed infectious period, the Weibull and log-logistic pairwise regression models produced slightly higher SAR estimates and wider confidence intervals than the exponential model. To compare goodness-of-fit among the parametric pairwise survival models, we used the Akaike Information Criterion (AIC). For the 4-day infectious period, the Weibull and log-logistic models had lower AICs than the exponential model. For the 6-day and 8-day infectious periods, the exponential model had the lowest AIC. The chain binomial and pairwise regression estimates are consistent with each other, but neither is consistent with the binomial estimates.

Fig 5 shows histograms of the simulated outbreak sizes in the LA households based on the four different SAR estimates that assume a 6-day infectious period. The binomial estimates predict outbreaks much larger than observed, but the chain binomial and pairwise estimates

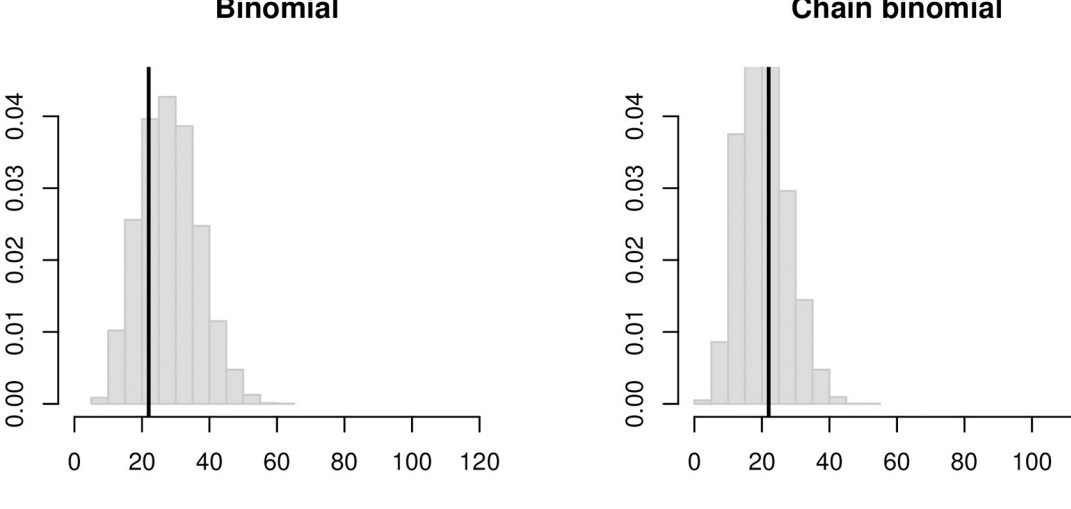

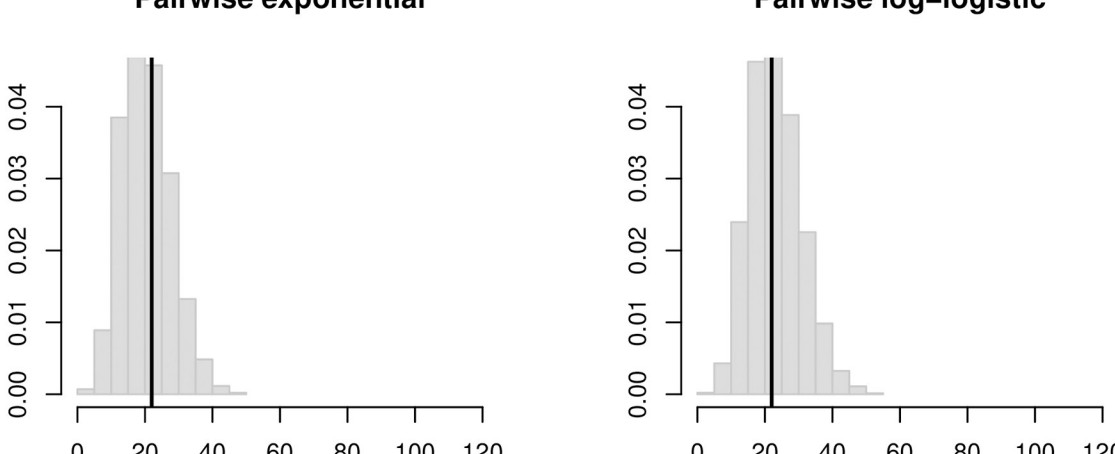

**Fig 6. Histogram of simulated final outbreak sizes in the LA households based on SAR estimates assuming a 4-day infectious period.** Vertical black lines indicate the observed final size of 22 cases.

predict outbreak size distributions centered near observed final sizes. Figs 6 and 7 show a similar pattern for estimates that assume 4-day and 8-day infectious periods, respectively. For the binomial estimates, the predicted outbreak sizes increase rapidly as the assumed infectious period gets longer. For the chain binomial and pairwise regression estimates, the predicted outbreak sizes increase much more slowly. To the extent that a true household SAR exists, it is almost certainly below the binomial estimates and closer to the chain binomial and pairwise regression estimates.

An important advantage of the longitudinal chain binomial and pairwise regression models is that they can estimate the SAR using the entire period of household observation. Table 4 shows point and interval estimates of the SAR based on the full data set collected by the LACDPH. As before, the chain binomial and pairwise exponential models produce nearly identical point and interval estimates. Using the full data set, the pairwise Weibull and log-

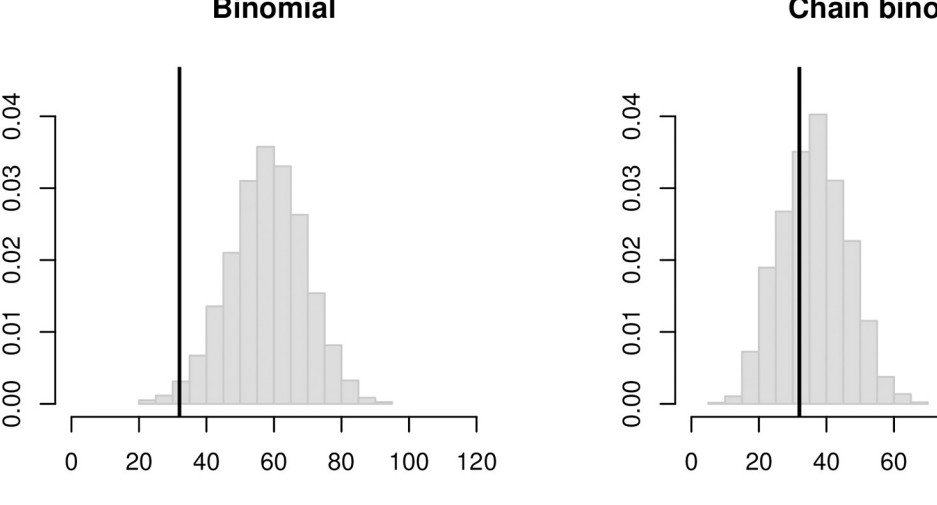

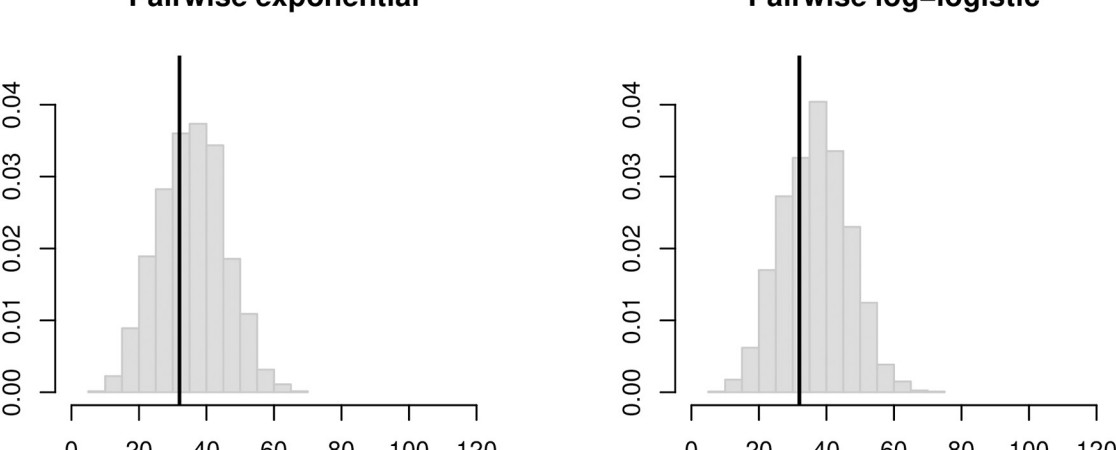

**Fig 7. Histogram of simulated final outbreak sizes in the LA households based on SAR estimates assuming an 8-day infectious period.** Vertical black lines indicate the observed final size of 32 cases.

logistic models produce point estimates closer to those of the one-parameter models than in Table 3, but their confidence intervals remain slightly wider. For the 6-day and 8-day assumed infectious periods, all four models produce lower point estimates of the SAR when using the full data set than when using only the possible second generation data. For the 4-day assumed infectious period, the point estimates from the full data are near the higher point estimates from the possible second-generation data. Fig 8 shows the distribution of outbreak sizes under the pairwise exponential estimate of the SAR assuming 6-, 4-, and 8-day infectious periods. The light gray histograms in the background show the distributions based on the point estimates from Table 3, which used the possible second generation data. In all three cases, there is a small but clear improvement in the predictive fit of the model when the full data set is used. Similar results were seen for the longitudinal chain binomial and pairwise Weibull and log-logistic regression models (see figures produced by S3 File).

**Table 4. Full-data estimates of the household SAR with 95% confidence limits and Akaike information criterion (AIC) for pairwise regression models.**

| Model | | Estimated SAR | | AIC |
|---|---|---|---|---|
| **6-day infectious period** | | | | |
| Longitudinal chain binomial | | 0.069 | (0.048, 0.096) | |
| Pairwise regression: | exponential | 0.068 | (0.048, 0.097) | 288.33 |
| | Weibull | 0.069 | (0.050, 0.111) | 289.74 |
| | log-logistic | 0.069 | (0.050, 0.111) | 289.54 |
| **4-day infectious period** | | | | |
| Longitudinal chain binomial | | 0.063 | (0.043, 0.089) | |
| Pairwise regression: | exponential | 0.062 | (0.043, 0.088) | 256.67 |
| | Weibull | 0.063 | (0.045, 0.108) | 254.30 |
| | log-logistic | 0.063 | (0.044, 0.105) | 254.24 |
| **8-day infectious period** | | | | |
| Longitudinal chain binomial | | 0.080 | (0.056, 0.108) | |
| Pairwise regression: | exponential | 0.079 | (0.056, 0.111) | 339.98 |
| | Weibull | 0.079 | (0.058, 0.122) | 341.97 |
| | log-logistic | 0.079 | (0.058, 0.119) | 341.66 |

## Discussion

Studies of disease transmission in households and other clearly-defined groups at risk of infection will always be one of the most effective means of obtaining critical information about routes of transmission, predictors of infectiousness and susceptibility, and the natural history of epidemic diseases [3, 8, 60]. Every author of the studies cited above has made an important contribution to infectious disease epidemiology and to public health. However, these studies should no longer be analyzed using binomial models. Even when the SAR is small, it is important to account for multiple generations of transmission. Unless these generations are clearly separated in time, a binomial estimate of the SAR will be biased upward and have a confidence interval with low coverage probability whether or not it is adjusted for clustering.

A binomial model can estimate the household FAR accurately if cluster-adjusted confidence intervals are used. However, the FAR was clearly defined in our simulations only because we made the following assumptions: (1) each household had at most one primary case, (2) all households were the same size, (3) susceptibles were not at risk of infection from outside the household after the occurrence of a primary case. In practice, these assumptions are extremely unlikely to hold. The LACDPH data had households with multiple primary cases, household sizes that varied from 2 to 20, an ongoing risk of infection from outside the household. Unlike the household FAR, the household SAR can be clearly defined under more realistic conditions.

Our simulation study of the SAR and FAR also assumed that individuals were identical in terms of infectiousness and susceptibility, which is extremely unlikely to hold in practice. In the LACDPH data, individuals varied in age, antiviral prophylaxis use, and other possible predictors of infectiousness and susceptibility. Some household studies have seen evidence of lower transmission intensity between individuals in larger households [21]. The chain binomial model and pairwise survival models allow the probability or hazard of transmission to depend on individual-level, pairwise, and household-level covariates [51, 52]. These covariate effects are estimated simultaneously, which is critical to preventing bias for contagious outcomes [61]. Accurate estimates of these effects can provide critical insight into the effectiveness of public health interventions such as handwashing, social distancing, antiviral prophylaxis or treatment, and vaccination.

### 4–day infectious period

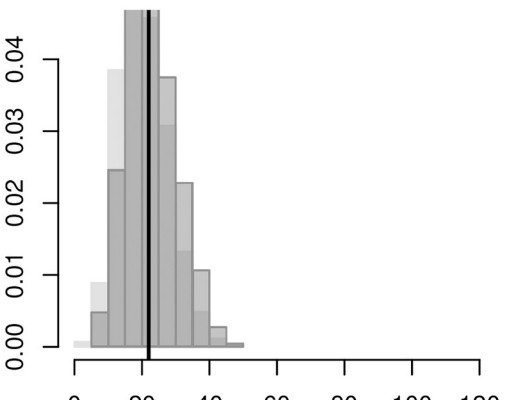

### 6–day infectious period

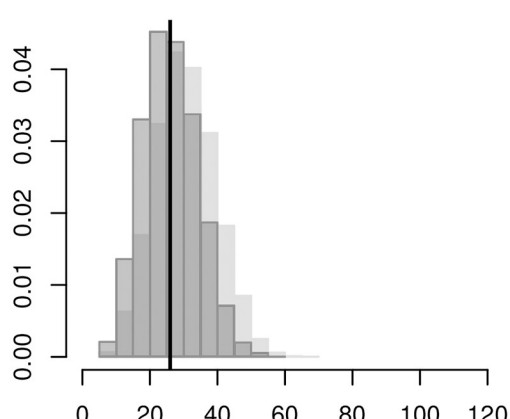

### 8–day infectious period

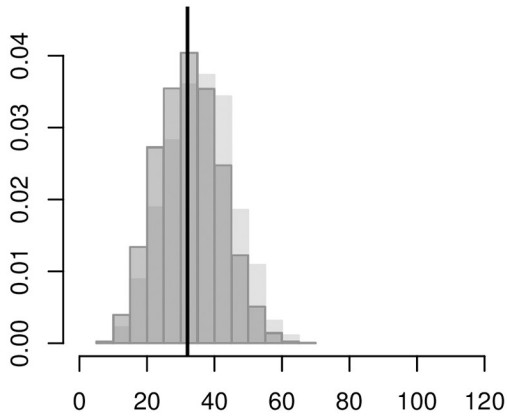

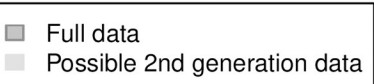

- Full data
- Possible 2nd generation data

**Fig 8. Histograms of simulated outbreak sizes based on pairwise exponential SAR estimates using the full data (dark gray) superimposed on the corresponding histograms from Figs 5–7 based on estimates using second generation data (light gray).** For each assumed infectious period, a vertical black line shows the observed final outbreak size.

The discrete-time chain binomial model [48] and pairwise survival models [49–51] require more detailed follow-up of each household than binomial models, but they can account accurately for delayed entry, loss to follow-up, and the risk of infection from outside the household. Most household studies already collect the data needed to use chain binomial and pairwise survival models for analysis. An additional advantage of these models is that data augmentation and Markov chain Monte Carlo (MCMC) can be used to fit them if there are undetected infections or if infection times cannot be determined precisely [62].

Whereas binomial models can be fit using almost any standard statistical package, the lack of available software has been a major obstacle to the adoption of statistical models of infectious disease transmission in household studies. Chain binomial models are available in the free and open source software package TranStat (www.cidid.org/transtat), which incorporates several advanced methods [63, 64] and has been used in analyses of influenza [20], Zika virus

[65], and Ebola virus [29]. Pairwise survival models are available in the free and open source `transtat` package for R, which was used to analyze the LA household data above. This package includes parametric models and semiparametric models [49–52, 57].

In the COVID-19 pandemic, there have been too few studies of SARS-CoV-2 transmission in households or other clearly-defined populations at risk of infection, leaving many questions about the intensity of transmission and the predictors of infectiousness and susceptibility unanswered [60]. This has forced public health decisions that affect millions of lives to be made under far greater uncertainty than there could or should have been. Household studies can provide critical scientific insights to guide public health interventions and policies. The results above show that replacing binomial models with chain binomial or pairwise survival models will help these studies contribute more effectively to the prevention and control of epidemics.

## Supporting information

**S1 File. R [56] (https://www.r-project.org/) code used to analyze the household outbreak simulations in section.** Produces Figs 1–4. Requires the following packages:

- gee [66] (https://cran.r-project.org/package=gee)

- reticulate [67] (https://rstudio.github.io/reticulate/)

- survival [68] (https://cran.r-project.org/package=survival)

- transtat [69] (https://github.com/ekenah/transtat)

Directions and package versions used for publication are in comments.
(R)

**S2 File. Python 3 [55] (https://www.python.org) functions called by S1 File.** Requires the following packages:

- NetworkX [70] (https://networkx.github.io)

- NumPy [71] and pandas [72] (https://www.scipy.org)

Directions and package versions used for publication are in comments.
(PY)

**S3 File. R [56] code used to analyze LACDPH household surveillance data in section.** Produces Tables 2 to 4 and Figs 5–8. In addition to the packages listed in S1 File, the following packages are required:

- MASS [73] (https://cran.r-project.org/package=MASS)

- sandwich [74] (https://cran.r-project.org/package=sandwich)

- stats4 (https://cran.r-project.org/package=stats4)

Directions and package versions used for publication are in comments.
(R)

**S4 File. Python 3 [55] functions called by S3 File.** Requires the NetworkX and pandas packages listed under S2 File: Directions and package versions used for publication are in comments.
(PY)

**S5 File. De-identified LACDPH household surveillance data in CSV format used by S3 File.** (CSV)

## Acknowledgments

The subtitle of the paper was inspired by Edsger Dijkstra's letter "Go To Statement Considered Harmful" (*Communications of the ACM* 11:147–148, 1968). Brit Oiulfstad, Dee Ann Bagwell, Brandon Dean, Laurene Mascola, and Elizabeth Bancroft of the Los Angeles County Department of Public Health (LACDPH) generously provided the household influenza surveillance data.

Disclaimer: The contribution of YS was completed prior to his Food and Drug Administration (FDA) employment. The content is solely the responsibility of the authors and does not represent the official views or policies of LACDPH, NIAID, NIGMS, NIH, or FDA.

## Author Contributions

**Conceptualization:** Eben Kenah.

**Data curation:** Eben Kenah.

**Formal analysis:** Yushuf Sharker, Eben Kenah.

**Funding acquisition:** Eben Kenah.

**Investigation:** Yushuf Sharker, Eben Kenah.

**Methodology:** Yushuf Sharker, Eben Kenah.

**Project administration:** Eben Kenah.

**Resources:** Eben Kenah.

**Software:** Yushuf Sharker, Eben Kenah.

**Supervision:** Eben Kenah.

**Validation:** Yushuf Sharker, Eben Kenah.

**Visualization:** Yushuf Sharker, Eben Kenah.

**Writing – original draft:** Yushuf Sharker, Eben Kenah.

**Writing – review & editing:** Yushuf Sharker, Eben Kenah.

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
