## [Decision Letter · Decision Letter 0]

8 Sep 2020

Dear Dr. Kenah,

Thank you very much for submitting your manuscript "Estimating and interpreting secondary attack risk: Binomial considered harmful" for consideration at PLOS Computational Biology. As with all papers reviewed by the journal, your manuscript was reviewed by members of the editorial board and by several independent reviewers. The reviewers appreciated the attention to an important topic. Based on the reviews, we are likely to accept this manuscript for publication, providing that you modify the manuscript according to the review recommendations.

Sincerely,

Cecile Viboud

Associate Editor

PLOS Computational Biology

Virginia Pitzer

Deputy Editor

PLOS Computational Biology

[LINK]

Reviewer's Responses to Questions

**Comments to the Authors:**

Reviewer #1: Please see attachment for detailed comments

Reviewer #2: This manuscript highlights the risk of using binomial models to estimate the secondary attack risk in clusters and highlights the importance of accounting for multiple generations of cases. This is a well-know problem for mathematical epidemiologists that, unfortunately, it is much less clear to more traditional epidemiologists. I believe this manuscript deals with a very important issue and provides a clear and accessible way to understand it for traditional epidemiologists. The theoretical framework looks solid to me, while the analysis of the LACDPH household data has a key flaw that should be addressed (see below).

Detailed comments

When analyzing the LACDPH household data, the authors are assuming that all influenza infections will result in an acute febrile respiratory illness. There are several studies showing that the probability of developing fever after influenza infection is <30% - I would like to point the authors to Carrat et al, Am J Epidemiol, 2008 and references therein. Therefore, it is very likely that the LACDPH missed more than half of the influenza infections. I agree with the authors’ choice to keep the transmission model as simple as possible and not include other factors such as age-specific susceptibility to infection, age-specific infectiousness, etc. However, the probability of developing fever is so important to proper interpreting the LACDPH household data that that cannot be neglected in the simulation analysis. As such all claims that some cases cannot be explained unless the re-importation of the infection to the household should be removed (e.g., line 279, 346). Also the definition of “late case” (line 276) should be revisited accordingly.

As for most infectious diseases, the duration of the infectious period of influenza is unknown and we have only rather indirect estimates of it. Therefore, I agree with the authors’ idea of exploring a wide range of values. However, I see two issues here. First, the list of explored values is way too large. Second, the distribution is clearly not uniform. We have a wide range of studies showing that the mean generation time of influenza is about 2-4 days. An infectious period of 12 days would be possible only if the transmission probability after the very first few days is extremely low. This should be clearly discussed and I suggest to use a more realistic (shorter) value for the infectious period in the baseline analysis and to decrease also the value of the upper bound as 12 days appear to be highly unrealistic.

I think it would be very interesting to look at the FAR by household size in the LACDPH and provide a comparison with the obtained modeling results. In fact, I fear that the model may fail the comparison with the data in this respect. If so, this should be clearly stated and acknowledged as a study limitation possibly linked to the many additional factors that are not included in the simple models used here.

Line 61-62. As stated before, the duration of the infectious period of influenza is unknown. The same applies also to the latent period. What we do know are the length of the generation time (mean roughly in the range 2-4 days) and of the incubation period (mean roughly in the range 1-2 days). I strongly recommend rephrasing this sentence in terms of incubation period and generation time that would better support the (correct) authors reasoning here.

Line 71-72, “so the binomial […] of the SAR”. I recommend dropping this part of the sentence.

Line 331. I suggest dropping the reproduction number from the list given here.

There are a few very minor English mistakes here and there (e.g., lines 47, 55, 57).

**Have all data underlying the figures and results presented in the manuscript been provided?**

Reviewer #1: Yes

Reviewer #2: Yes

PLOS authors have the option to publish the peer review history of their article (what does this mean?). If published, this will include your full peer review and any attached files.

Reviewer #1: No

Reviewer #2: **Yes: **Marco Ajelli
---

## [Editor Report · Decision Letter 1]

2 Dec 2020

Dear Dr. Kenah,

We are pleased to inform you that your manuscript 'Estimating and interpreting secondary attack risk: Binomial considered biased' has been provisionally accepted for publication in PLOS Computational Biology.

Best regards,

Cecile Viboud

Associate Editor

PLOS Computational Biology

Virginia Pitzer

Deputy Editor

PLOS Computational Biology

---

## [Editor Report · Acceptance letter]

15 Jan 2021

PCOMPBIOL-D-20-01278R1 

Estimating and interpreting secondary attack risk: Binomial considered biased

Dear Dr Kenah,

I am pleased to inform you that your manuscript has been formally accepted for publication in PLOS Computational Biology. Your manuscript is now with our production department and you will be notified of the publication date in due course.

With kind regards,

Jutka Oroszlan
